# ACTION-SUFFICIENT STATE REPRESENTATION LEARNING FOR CONTROL WITH STRUCTURAL CONSTRAINTS

**Biwei Huang**[*1]**, Chaochao Lu**[*2,3]**, Leqi Liu**[1]**, Jośe Miguel Hernández-Lobato**[2,4]**,
Clark Glymour**[1]**, Bernhard Schölkopf** [3]**, Kun Zhang**[1]

## ABSTRACT

Perceived signals in real-world scenarios are usually high-dimensional and noisy, and finding and using their representation that contains essential and sufficient information required by downstream decision-making tasks will help improve computational efficiency and generalization ability in the tasks. In this paper, we focus on partially observable environments and propose to learn a minimal set of state representations that capture sufficient information for decision-making, termed *Action-Sufficient state Representations* (ASRs). We build a generative environment model for the structural relationships among variables in the system and present a principled way to characterize ASRs based on structural constraints and the goal of maximizing cumulative reward in policy learning. We then develop a structured sequential Variational Auto-Encoder to estimate the environment model and extract ASRs. Our empirical results on CarRacing and VizDoom demonstrate a clear advantage of learning and using ASRs for policy learning. Moreover, the estimated environment model and ASRs allow learning behaviors from imagined outcomes in the compact latent space to improve sample efficiency.

## 1 INTRODUCTION

State-of-the-art reinforcement learning (RL) algorithms leveraging deep neural networks are usually data hungry and lack interpretability. For example, to attain expert-level performance on tasks such as chess or Atari games, deep RL systems usually require many orders of magnitude more training data than human experts (Tsividis et al., 2017). One of the reasons is that our perceived signals in real-world scenarios, e.g., images, are usually high-dimensional and may contain much irrelevant information for decision-making of the task at hand. This makes it difficult and expensive for an agent to directly learn optimal policies from raw observational data. Fortunately, the underlying states that directly guide decision-making could be much lower-dimensional (Schölkopf, 2019; Bengio, 2019). One example is that when crossing the street, our decision on when to cross relies on the traffic lights. The useful state of traffic lights (e.g., its color) can be represented by a single binary variable, while the perceived image is high-dimensional. It is essential to extract and exploit such lower-dimensional states to improve the efficiency and interpretability of the decision-making process.

Recently, representation learning algorithms have been designed to learn abstract features from high-dimensional and noisy observations. Exploiting the abstract representations, instead of the raw data, has been shown to perform subsequent decision-making more efficiently (Lesort et al., 2018). Representative methods along this line include deep Kalman filters (Krishnan et al., 2015), deep variational Bayes filters (Karl et al., 2016), world models (Ha & Schmidhuber, 2018), PlaNet (Hafner et al., 2018), DeepMDP (Gelada et al., 2019), stochastic latent actor-critic (Lee et al., 2019), SimPLe (Kaiser et al., 2019), Bisimulation-based methods (Zhang et al., 2021), Dreamer (Hafner et al., 2019; 2020), and others (Srinivas et al., 2020; Shu et al., 2020). Moreover, if we can properly model and estimate the underlying transition dynamics, then we can perform model-based RL or planning, which can effectively reduce interactions with the environment (Ha & Schmidhuber, 2018; Hafner et al., 2018; 2019; 2020).

Despite the effectiveness of the above approaches to learning abstract features, current approaches usually fail to take into account whether the extracted state representations are sufficient and necessary for downstream policy learning. State representations that contain insufficient information may lead to sub-optimal policies, while those with redundant information may require more samples and more complex models for training. We address this problem by modeling the generative process and selection procedure induced by reward maximization; by considering a generative environment model involving observed states, state-transition dynamics, and rewards, and explicitly characterizing

[1]Carnegie Mellon University, [2]University of Cambridge, [3]MPI for Intelligent Systems, [4]The Alan Turing Institute, [*]Equal Contribution, Correspondence at `biweih@andrew.cmu.edu`, `cl641@cam.ac.uk`.

structural relationships among variables in the RL system, we propose a principled approach to learning minimal sufficient state representations. We show that only the state dimensions that have direct or indirect edges to the reward variable are essential and should be considered for decision making. Furthermore, they can be learned by maximizing their ability to predict the action, given that the cumulative reward is included in the prediction model, while at the same time achieving their minimality w.r.t. the mutual information with observations as well as their dimensionality. The contributions of this paper are summarized as follows:

- We construct a generative environment model, which includes the observation function, transition dynamics, and reward function, and explicitly characterizes structural relationships among variables in the RL system.
- We characterize a minimal sufficient set of state representations, termed Action-Sufficient state Representations (ASRs), for the downstream policy learning by making use of structural constraints and the goal of maximizing cumulative reward in policy learning.
- We develop Structured Sequential Variational Auto-Encoder (SS-VAE), which explicitly encodes structural relationships among variables, for reliable identification of ASRs.
- Policy learning can be done separately from representation learning, and the policy function only relies on a set of low-dimensional state representations, which improve both model and sample efficiency. Moreover, the estimated environment model and ASRs allow learning behaviors from imagined outcomes in the compact latent space, which effectively reduce risky explorations.

## 2 ENVIRONMENT MODEL WITH STRUCTURAL CONSTRAINTS

To characterize a set of minimal sufficient state representations for downstream policy learning, we first formulate a generative environment model in partially observable Markov decision process (POMDP), and then show how to explicitly embed structural constraints over variables in the RL system and leverage them.

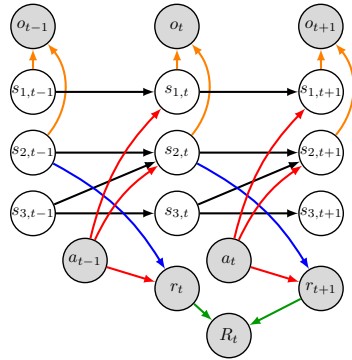

Figure 1: A graphical illustration of the generative environment model. Grey nodes denote observed variables and white nodes represent unobserved variables. Here, $a_{t-1}$ does not have an edge to $s_{3,t}$, and only $s_{2,t-1}$ and $s_{3,t-1}$ have edges to $r_t$. We take into account the structural relationships among different dimensions of latent states $\vec{s}_t$.

Suppose we have sequences of observations $\{\langle o_t, a_t, r_t \rangle\}_{t=1}^T$, where $o_t \in \mathcal{O}$ denotes perceived signals at time $t$, such as high-dimensional images, with $\mathcal{O}$ being the observation space, $a_t \in \mathcal{A}$ is the performed action with $\mathcal{A}$ being the action space, and $r_t \in \mathcal{R}$ represents the reward variable with $\mathcal{R}$ being the reward space. We denote the underlying states, which are latent, by $\vec{s}_t \in \mathcal{S}$, with $\mathcal{S}$ being the state space. We describe the generating process of the environment model as follows:

$$\begin{cases} o_t = f(\vec{s}_t, e_t), \\ r_t = g(\vec{s}_{t-1}, a_{t-1}, \epsilon_t), \\ \vec{s}_t = h(\vec{s}_{t-1}, a_{t-1}, \eta_t), \end{cases} \quad (1)$$

where $f$, $g$, and $h$ represent the observation function, reward function, and transition dynamics, respectively, and $e_t$, $\epsilon_t$, and $\eta_t$ are corresponding independent and identically distributed (i.i.d.) random noises. The latent states $\vec{s}_t$ form an MDP: given $\vec{s}_{t-1}$ and $a_{t-1}$, $\vec{s}_t$ are independent of states and actions before $t-1$. Moreover, the action $a_{t-1}$ directly influences latent states $\vec{s}_t$, instead of perceived signals $o_t$, and the reward is determined by the latent states (and the action) as well. The perceived signals $o_t$ are generated from the underlying states $\vec{s}_t$, contaminated by random noise $e_t$. We also consider noise $\epsilon_t$ in the reward function to capture unobserved factors that may affect the reward, as well as measurement noise.

It is commonplace that the action variable $a_{t-1}$ may not influence every dimension of $\vec{s}_t$, and the reward $r_t$ may not be influenced by every dimension of $\vec{s}_{t-1}$ as well, and furthermore there are structural relationships among different dimensions of $\vec{s}_t$. Fig. 1 gives an illustrative graphical representation, where $s_{3,t-1}$ influences $s_{2,t}$, $a_{t-1}$ does not have an edge to $s_{3,t}$, and among the states, only $s_{2,t-1}$ and $s_{3,t-1}$ have edges to $r_t$. We use $R_t = \sum_{\tau=t}^\infty \gamma^{\tau-t} r_\tau$ to denote the discounted cumulative reward starting from time $t$, where $\gamma \in [0, 1]$ is the discounted factor that determines how much immediate rewards are favored over more distant rewards.

To reflect such constraints, we explicitly encode the graph structure over variables, including the structure over different dimensions of $\vec{s}$ and the structures from $a_{t-1}$ to $\vec{s}_t$, $\vec{s}_{t-1}$ to $r_t$, and $\vec{s}_t$ to $o_t$.

Accordingly, we re-formulate Eq. 1 as follows:

$$\begin{cases} o_t = f(D_{\vec{s} \to o} \odot \vec{s}_t, e_t), \\ r_t = g(D_{\vec{s} \to r} \odot \vec{s}_{t-1}, D_{a \to r} \odot a_{t-1}, \epsilon_t), \\ s_{i,t} = h_i(D_{\vec{s}(\cdot,i)} \odot \vec{s}_{t-1}, D_{a \to \vec{s}(\cdot,i)} \odot a_{t-1}, \eta_{i,t}), \end{cases} \quad (2)$$

for $i = 1, \cdots, d$, where $\vec{s}_t = (s_{1,t}, \cdots, s_{d,t})^\top$, $\odot$ denotes element-wise product, and $D_{(\cdot)}$ are binary matrices indicating the graph structure over variables. Specifically, $D_{\vec{s} \to o} \in \{0,1\}^{d \times 1}$ represents the graph structure from $d$-dimensional $\vec{s}_t$ to $o_t$, $D_{\vec{s} \to r} \in \{0,1\}^{d \times 1}$ the structure from $\vec{s}_{t-1}$ to the reward variable $r_t$, $D_{a \to r} \in \{0,1\}$ the structure from the action variable $a_{t-1}$ to the reward variable $r_t$, $D_{\vec{s}} \in \{0,1\}^{d \times d}$ denotes the graph structure from $d$-dimensional $\vec{s}_{t-1}$ to $d$-dimensional $\vec{s}_t$ and $D_{\vec{s}(\cdot,i)}$ is its $i$-th column, and $D_{a \to \vec{s}} \in \{0,1\}^{1 \times d}$ corresponds to the graph structure from $a_{t-1}$ to $\vec{s}_t$ with $D_{a \to \vec{s}(\cdot,i)}$ representing its $i$-th column. For example, $D_{\vec{s}(j,i)} = 0$ means that there is no edge from $s_{j,t-1}$ to $s_{i,t}$. Here, we assume that the environment model, as well as the structural constraints, is invariant across time instance $t$.

## 2.1 Minimal Sufficient State Representations

Given observational sequences $\{\langle o_t, a_t, r_t \rangle\}_{t=1}^T$, we aim to learn minimal sufficient state representations for the downstream policy learning. In the following, we first characterize the state dimensions that are indispensable for policy learning, when the environment model, including structural relationships, is given. Then we provide criteria to achieve sufficiency and minimality of the estimated state representations, when only $\{\langle o_t, a_t, r_t \rangle\}_{t=1}^T$, but not the environment model, is given.

**Finding minimal sufficient state dimensions with a given environment model.** RL agents learn to choose appropriate actions according to the current state vector $\vec{s}_t$ to maximize the cumulative reward, in which some dimensions may be redundant for policy learning. Then how can we identify a minimal subset of state dimensions that are sufficient to choose optimal actions? We call such state dimensions *Action-Sufficient state Representations (ASRs)*, and denote it by $\vec{s}_t^{\text{ASR}}$. Below, we give the graphical condition that ASRs are expected to satisfy.

**Proposition 1.** *Given the graphical representation corresponding to the environment model, such as the representation in Fig. 1, which is assumed to be Markov and faithful to the measured data, each dimension in $\vec{s}_t^{ASR}$ has a direct or indirect edge to $r_{t+\tau}$, for $\tau > 0$.*

Proposition 1 can be shown based on the global Markov condition and the faithfulness assumption (Pearl, 2000; Spirtes et al., 1993). A proof is given in the appendix. If $s_{i,t}$ has a (direct or indirect) edge to $r_{t+\tau}$, the action variable $a_t$ is dependent on $s_{i,t}$ when the cumulative reward is maximized, which can be thought of as a data selection procedure depending on the cumulative reward, and thus $s_{i,t}$ is needed for action determination. On the other hand, if $s_{i,t}$ does not have an edge to $r_{t+\tau}$, then the action variable $a_t$ is (conditionally) independent on $s_{i,t}$, and thus $s_{i,t}$ is not needed for action determination. According to Proposition 1, for the graph in Fig. 1, we have $\vec{s}_t^{\text{ASR}} = (s_{2,t}, s_{3,t})^\top$. That is, we only need $(s_{2,t}, s_{3,t})^\top$, instead of $\vec{s}_t$, for the downstream policy learning.

**Minimal sufficient state representation learning from observed sequences.** In practice, we usually do not have access to the latent states or the environment model, but instead only the observed sequences $\{\langle o_t, a_t, r_t \rangle\}_{t=1}^T$. Then how can we learn the ASRs from the raw high-dimensional inputs such as images? We denote by $\tilde{\vec{s}}_t$ the estimated whole latent state representations and $\tilde{\vec{s}}_t^{\text{ASR}} \subseteq \tilde{\vec{s}}_t$ the estimated minimal sufficient state representations for policy learning.

We collected the data, that are used to learn the environment model and ASRs, with random actions. As discussed above, action and ASRs are dependent given the cumulative reward—this is a type of dependence relationship induced by selection on the effect (reward). We can then learn the ASRs by maximizing their dependence with the action given the cumulative reward, $I(\tilde{\vec{s}}_t^{\text{ASR}}; a_t \mid R_{t+1})$, where $I$ denotes mutual information. Since $I(\tilde{\vec{s}}_t^{\text{ASR}}; a_t \mid R_{t+1}) = H(a_t \mid R_{t+1}) - H(a_t \mid \tilde{\vec{s}}_t^{\text{ASR}}, R_{t+1})$, where $H(\cdot)$ denotes the conditional entropy, and the first term $H(a_t \mid R_{t+1})$ does not contain ASRs, we can estimate ASRs by minimizing $H(a_t \mid \tilde{\vec{s}}_t^{\text{ASR}}, R_{t+1})$, with

$$H(a_t \mid \tilde{\vec{s}}_t^{\text{ASR}}, R_{t+1}) = -\sum_{t=1}^T \mathbb{E}_{q_{\phi,\alpha}}\{\log p_\alpha(a_t \mid \tilde{\vec{s}}_t^{\text{ASR}}, R_{t+1})\} = -\sum_{t=1}^T \mathbb{E}_{q_{\phi,\alpha}}\{\log p_\alpha(a_t \mid \tilde{D}^{\text{ASR}} \odot \tilde{\vec{s}}_t, R_{t+1})\},$$

where $p_\alpha$ denote the probabilistic predictive model of $a_t$ with parameters $\alpha$, $q_{\phi,\alpha}$ is the joint distribution over $\tilde{\vec{s}}_t$ and $a_t$ with $q_{\phi,\alpha} = q_\phi p_\alpha$, and $q_\phi(\tilde{\vec{s}}_t \mid \tilde{\vec{s}}_{t-1}, \mathbf{y}_{1:t}, a_{1:t-1})$ is the probabilistic inference model of $\tilde{\vec{s}}_t$ with parameters $\phi$ and $\mathbf{y}_t = (o_t^T, r_t^T)$, and $\tilde{D}^{\text{ASR}} \in \{0,1\}^{d \times 1}$ is a binary vector indicating which dimensions of $\tilde{\vec{s}}_t$ are in $\tilde{\vec{s}}_t^{\text{ASR}}$, so $\tilde{D}^{\text{ASR}} \odot \tilde{\vec{s}}_t$ gives ASRs $\tilde{\vec{s}}_t^{\text{ASR}}$.

Moreover, we achieve minimality of the representation by minimizing conditional mutual information between observed high-dimensional signals $\mathbf{y}_t$ and the ASR $\tilde{\tilde{s}}_t^{\text{ASR}}$ at time $t$ given data at previous time instances, and meanwhile minimizing the dimensionality of ASRs with sparsity constraints:

$$\lambda_1 \sum_{t=2}^{T} I(\mathbf{y}_t; \tilde{\tilde{s}}_t^{\text{ASR}} | \mathbf{y}_{1:t-1}, a_{1:t-1}, \tilde{\tilde{s}}_{t-1}) + \lambda_2 \|\check{D}^{\text{ASR}}\|_1,$$

where the conditional mutual information can be upper bound by a KL-divergence:

$$I(\mathbf{y}_t; \tilde{\tilde{s}}_t^{\text{ASR}} | \mathbf{y}_{1:t-1}, a_{1:t-1}, \tilde{\tilde{s}}_{t-1}) \leq \int q_\phi(\tilde{\tilde{s}}_t^{\text{ASR}} | \tilde{\tilde{s}}_{t-1}, \mathbf{y}_{1:t}, a_{1:t-1}) \log \frac{q_\phi(\tilde{\tilde{s}}_t^{\text{ASR}} | \tilde{\tilde{s}}_{t-1}, \mathbf{y}_{1:t-1})}{p_\gamma(\tilde{\tilde{s}}_t^{\text{ASR}} | \tilde{\tilde{s}}_{t-1}, a_{t-1}; D_{\vec{s}}, D_{a \to \vec{s}})} \, d\tilde{\tilde{s}}_t^{\text{ASR}}$$
$$= \text{KL}\big(q_\phi(\tilde{\tilde{s}}_t^{\text{ASR}} | \tilde{\tilde{s}}_{t-1}, \mathbf{y}_{1:t}, a_{1:t-1}) \| p_\gamma(\tilde{\tilde{s}}_t^{\text{ASR}} | \tilde{\tilde{s}}_{t-1}, a_{t-1}; D_{\vec{s}}, D_{a \to \vec{s}})\big),$$

with $p_\gamma$ being the transition dynamics of $\tilde{\tilde{s}}_t$ with parameters $\gamma$.

Furthermore, Proposition 1 shows that given the (estimated) environment model, only those state dimensions that have a direct or indirect edge to the reward variable are the ASRs. In our learning procedure, we also take into account the relationship between the learned states $\tilde{\tilde{s}}_t$ and the reward, and leverage such structural constraints for learning the ASRs. Denote by $\check{D}^{ASR} \in \{0,1\}^{\check{d} \times 1}$ a binary vector indicating whether the corresponding state dimension in $\tilde{\tilde{s}}_t$ has a direct or indirect edge to the reward variable. Consequently, we enforce the similarity between $\check{D}^{ASR}$ and $\tilde{D}^{ASR}$ by adding an $L_1$ norm on $\check{D}^{ASR} - \tilde{D}^{ASR}$. Therefore, the ASRs can be learned by maximizing

$$\mathcal{L}^{\text{min \& suff}} = \lambda_3 \underbrace{\sum_{t=1}^{T} \mathbb{E}_{q_{\phi,\alpha}} \big\{ \log p_\alpha(a_t | \tilde{D}^{ASR} \odot \tilde{\tilde{s}}_t, R_{t+1}) \big\}}_{\text{Sufficiency}} - \lambda_4 \|\check{D}_{\text{ASR}} - \tilde{D}_{\text{ASR}}\|_1$$
$$- \lambda_1 \sum_{t=2}^{T} \underbrace{\text{KL}\big(q_\phi(\tilde{D}^{ASR} \odot \tilde{\tilde{s}}_t | \tilde{\tilde{s}}_{t-1}, \mathbf{y}_{1:t-1}) \| p_\gamma(\tilde{D}^{ASR} \odot \tilde{\tilde{s}}_t | \tilde{\tilde{s}}_{t-1}, a_{t-1}; D_{\vec{s}}, D_{a \to \vec{s}})\big)}_{\text{Minimality}} - \lambda_2 \|\tilde{D}^{ASR}\|_1, \qquad (3)$$

where $\lambda$'s are regularization terms, and note that $\check{D}^{ASR}$ can be directly derived from the estimated structural matrices $D_{a \to r}$ and $D_{\vec{s}(\cdot,i)}$. The constraint in Eq. 3 provides a principled way to achieve minimal sufficient state representations, related to the information bottleneck method (Tishby et al., 1999). Notice that it is just part of the objective function to maximize, and it will be involved in the complete objective function in Eq. 4 to learn the whole environment model.

# 3 STRUCTURED SEQUENTIAL VAE FOR THE ESTIMATION OF ASRS

We focus on estimation procedures for the environment model and ASRs in the nonlinear case. It is worth noting that in the linear cases, we can further give the identifiability result of the model, which is left in Appendix C. To handle general nonlinear cases with the generative process given in Eq. 2, we develop a Structured Sequential VAE (SS-VAE) to learn the model (including structural constraints) and infer latent state representations $\tilde{\tilde{s}}_t$ and ASRs $\tilde{\tilde{s}}_t^{\text{ASR}}$, with the input $\{\langle o_t, a_t, r_t \rangle\}_{t=1}^{T}$. Specifically, the latent state dimensions are organized with structures, captured by $D_{\vec{s}}$, to achieve conditional independence. The structural relationships over perceived signals, latent states, the action variable, and the reward variable are also embedded as free parameters (i.e., $D_{\vec{s} \to o}, D_{\vec{s} \to r}, D_{a \to r}, D_{a \to \vec{s}}$) into SS-VAE. Moreover, we aim to learn state representations $\tilde{\tilde{s}}_t$ and ASRs $\tilde{\tilde{s}}_t^{\text{ASR}}$ that satisfy the following properties: (*i*) $\tilde{\tilde{s}}_t$ should capture sufficient information of observations $o_t$, $r_t$, and $a_t$, that is, it should be enough to enable reconstruction. (*ii*) The state representations should allow for accurate predictions of the next state and also the next observation. (*iii*) The transition dynamics should follow an MDP. (*iv*) $\tilde{\tilde{s}}_t^{\text{ASR}}$ are minimal sufficient state representations for the downstream policy learning. Let $\mathbf{y}_{1:T} = \{(o_t^\top, r_t^\top)^\top\}_{t=1}^{T}$. To achieve these properties, we maximize the following objective:

$$\mathcal{L}(\mathbf{y}_{1:T}; (\theta, \phi, \gamma, \alpha, D_{(\cdot)})) = \sum_{t=1}^{T-2} \mathbb{E}_{q_\phi} \big\{ \underbrace{\log p_\theta(o_t | \tilde{\tilde{s}}_t; D_{\vec{s} \to o}) + \log p_\theta(r_{t+1} | \tilde{\tilde{s}}_t, a_t; D_{\vec{s} \to r}, D_{a \to r})}_{\text{Reconstruction}}$$
$$+ \underbrace{\log p_\theta(o_{t+1} | \tilde{\tilde{s}}_t) + \log p_\theta(r_{t+2} | \tilde{\tilde{s}}_t, a_{t+1})}_{\text{Prediction}} \big\} + \lambda_3 \sum_{t=1}^{T} \mathbb{E}_{q_{\phi,\alpha}} \big\{ \underbrace{\log p_\alpha(a_t | \tilde{\tilde{s}}_t, R_{t+1}; \tilde{D}_{\text{ASR}})}_{\text{Sufficiency}} \big\}$$
$$- \lambda_1 \sum_{t=2}^{T} \underbrace{\text{KL}\big(q_\phi(\tilde{\tilde{s}}_t | \tilde{\tilde{s}}_{t-1}, \mathbf{y}_{1:t}, a_{1:t-1}) \| \underbrace{p_\gamma(\tilde{\tilde{s}}_t | \tilde{\tilde{s}}_{t-1}, a_{t-1}; D_{\vec{s}}, D_{a \to \vec{s}})}_{\text{Transition}}\big)}_{\text{Conditional disentanglement \& Minimality}} - \lambda_2 \|\tilde{D}_{\text{ASR}}\|_1 \qquad (4)$$
$$- \underbrace{\big(\lambda_5 \|D_{\vec{s} \to o}\|_1 + \lambda_6 \|D_{\vec{s} \to r}\|_1 + \lambda_7 \|D_{\vec{s}}\|_1 + \lambda_8 \|D_{a \to \vec{s}}\|_1 + \lambda_4 \|\check{D}_{\text{ASR}} - \tilde{D}_{\text{ASR}}\|_1\big)}_{\text{Sparsity}}.$$

We denote by $p_\theta$ the generative model with parameters $\theta$ and structural constraints $D_{(\cdot)}$, $q_\phi$ the inference model with parameters $\phi$, $p_\gamma$ the transition dynamics, and $p_\alpha$ action prediction with ASRs given the cumulative reward $R_{t+1}$. Each factor in $p_\gamma$ and $q_\phi$ is modeled with a mixture of Gaussians

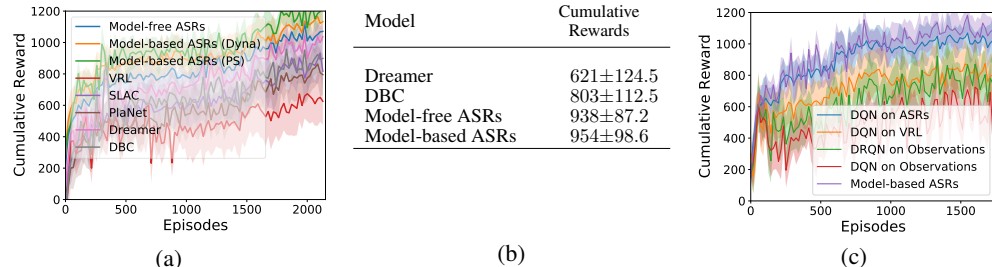

Figure 2: (a) Cumulative rewards of model-based ASRs, model-free ASRs, VRL, SLAC, PlaNet, DBC and Dreamer evaluated on CarRacing. (b) Comparisons with Dreamer and DBC on CarRacing with natural video distractors, after 2000 training episodes. (c) Comparing ASRs with SOTA methods on VizDoom.

(MoGs), to approximate a wide class of continuous distributions. The details of each component in the above objective and their corresponding neural network architecture are given in Appendix D.

## 4 POLICY LEARNING WITH ASRS

After estimating the generative environment model, we are ready to learn the optimal policy, where the policy function only depends on low-dimensional ASRs, instead of high-dimensional images. The entire procedure roughly contains the following three parts: (1) data collection with a random policy, (2) environment model estimation (with details in Section 3), and (3) policy learning with ASRs. Notably, the generative environment model is fixed, regardless of the behavior policy that is used to generate the data, and after learning the environment model, as well as the inference model for ASRs, our framework is flexible for both model-free and model-based policy learning. The details of the algorithms for policy learning is left in Appendix F.

## 5 EXPERIMENTS

To evaluate the proposed approach, we follow Ha & Schmidhuber (2018) to conduct experiments on both CarRacing (Klimov, 2016) and VizDoom (Kempka et al., 2016). Without stated otherwise, all results were averaged across five random seeds, with standard deviation. Due to space limit, we only highlight some key results and refer readers to Appendix G for more details.

We conduct a series of experiments on CarRacing to empirically show that the low-dimensional ASRs significantly improve the policy learning performance in terms of both efficiency and efficacy. (1) **Analysis of ASRs (Fig. 7 in Appendix H).** To demonstrate the structures over observed frames, latent states, actions, and rewards, we visualized the learned $D_{\vec{s} \to o}$, $D_{\vec{s} \to r}$, $D_{\vec{s}}$, and $D_{a \to \vec{s}}$. Compared to the original 32-dim latent states, ASRs have only 21 dimensions. (2) **Comparison Between Model-Free and Model-Based ASRs (Fig. 2a).** We applied both model-free (DDPG) (Lillicrap et al., 2015) and model-based (Dyna and Prioritized Sweeping) algorithms (Sutton, 1990) to ASRs (with 21-dims). (3) **Comparison with VRL, SLAC, PlaNet, DBC, and Dreamer (Fig. 2a).** We also compared the proposed framework of policy learning with ASRs (with 21-dims) with a) the same learning strategy but with vanilla representation learning (VRL, implemented without the components for minimal sufficient state representations as in Eq. 3), b) SLAC (Lee et al., 2019), c) PlaNet (Hafner et al., 2018), d) DBC (Zhang et al., 2021), and e) Dreamer (Hafner et al., 2019). For a fair comparison, the latent dimensions of VRL, PlaNet, SLAC, DBC and Dreamer are set to 21. (4) **Comparison with Dreamer and DBC with Background Distraction (Fig. 2b).** We further compared ASRs (with 21-dims) with Dreamer and DBC when there are natural video distractors (Zhang et al., 2021). (5) **Ablation Study (Fig. 8 in Appendix H).** We further performed ablation studies on latent dynamics prediction.

We also applied the proposed method to VizDoom *take cover* scenario (Kempka et al., 2016). Considering that in the *take over* scenario the action space is discrete, we applied the widely used DQN (Mnih et al., 2013) on ASRs for policy learning. In addition to the comparisons with VRL (as in CarRacing) and DQN on raw observations, we further compared with another common approach to POMDPs: DRQN (Hausknecht & Stone, 2015). As shown in Fig. 2c, DQN on ASRs achieve a much better performance than all other comparisons.

## 6 CONCLUSION AND RELATED WORK

In this paper, we develop a principled framework to characterize a minimal set of state representations that suffice for policy learning, by making use of structural constraints and action predictions. Accordingly, we propose SS-VAE to reliably extract such a set of state representations from raw observations. The estimated environment model and ASRs allow learning behaviors from imagined outcomes in the compact latent space, which effectively reduce sample complexity and possibly risky interactions with the environment. We discuss the related work in Appendix J.

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

# A    PROOF OF PROPOSITION 1

We first give the definitions of the Markov condition and the faithfulness assumption, which will be used in the proof.

**Definition 1** (Global Markov Condition (Spirtes et al., 1993; Pearl, 2000))**.** *The distribution $p$ over $\mathbf{V}$ satisfies the global Markov property on graph $G$ if for any partition $(A, B, C)$ such that $B$ d-separates $A$ from $C$,*

$$p(A, C|B) = p(A|B)p(C|B).$$

**Definition 2** (Faithfulness Assumption (Spirtes et al., 1993; Pearl, 2000))**.** *There are no independencies between variables that are not entailed by the Markov Condition.*

Below, we give the proof of Proposition 1.

*Proof.* We first show that if $s_{i,t} \in \vec{s}_t^{\text{ASR}}$, then it has a direct or indirect edge to $r_{t+\tau}$.

We prove it by contradiction. Suppose that $s_{i,t}$ does not have a direct or indirect edge to $r_{t+\tau}$. According to the Markov assumption, $s_{i,t}$ is independent of $a_t$ given $R$. Hence, $s_{i,t}$ is not necessary for decision making, and thus $s_{i,t}$ is not a dimension in $\vec{s}_t^{\text{ASR}}$, which contradicts to the assumption. Since we have a contradiction, it must be that $s_{i,t}$ has a direct or indirect edge to $r_{t+\tau}$.

We next show that if $s_{i,t}$ has a direct or indirect edge to $r_{t+\tau}$, then $s_{i,t} \in \vec{s}_t^{\text{ASR}}$.

Similarly, by contradiction suppose that $s_{i,t}$ is not a dimension in $\vec{s}_t^{\text{ASR}}$. It means that $s_{i,t}$ is independent on $a_t$ given $R$ and some other variables. Then according to the faithfulness assumption, $s_{i,t}$ does not have a direct or indirect edge to $r_{t+\tau}$, which contradicts to the assumption. $\square$

# B    MINIMALITY OF THE REPRESENTATION

In this section, we give the detailed derivation of the minimality of the state representation given in Section 2.1.

We achieve minimality of the representation by minimizing conditional mutual information between observed high-dimensional signals $\mathbf{y}_t$, where $\mathbf{y}_t = \{o_t^T, r_t^T\}$, and the ASR $\tilde{\vec{s}}_t^{\text{ASR}}$ at time $t$ given data at previous time instances, and meanwhile minimizing the dimensionality of ASRs with sparsity constraints:

$$\lambda_1 \sum_{t=2}^{T} I(\mathbf{y}_t; \tilde{\vec{s}}_t^{\text{ASR}} | \mathbf{y}_{1:t-1}, a_{1:t-1}, \tilde{\vec{s}}_{t-1}) + \lambda_2 \|\tilde{D}^{\text{ASR}}\|_1.$$

Note that in the above conditional mutual information, we need to conditional on the previous states $\tilde{\vec{s}}_{t-1}$, instead of $\tilde{\vec{s}}_{t-1}^{\text{ASR}}$, which two give different conditional mutual information. It can be shown by contradiction. Suppose $I(\mathbf{y}_t; \tilde{\vec{s}}_t^{\text{ASR}} | \mathbf{y}_{1:t-1}, a_{1:t-1}, \tilde{\vec{s}}_{t-1}) = I(\mathbf{y}_t; \tilde{\vec{s}}_t^{\text{ASR}} | \mathbf{y}_{1:t-1}, a_{1:t-1}, \tilde{\vec{s}}_{t-1}^{\text{ASR}})$, and denote $\tilde{\vec{s}}^C = \tilde{\vec{s}} \backslash \tilde{\vec{s}}^{\text{ASR}}$. Then the equivalence implies that $\tilde{\vec{s}}_{t-1}^C$ is independent of $o_t$ (where $o_t \in \mathbf{y}_t$) given $\{\mathbf{y}_{1:t-1}, a_{1:t-1}, \tilde{\vec{s}}_{t-1}^{\text{ASR}}\}$. It is obviously violated for the example given in Fig. 1, where $\tilde{\vec{s}}^C = s_1$ and $\tilde{\vec{s}}^{\text{ASR}} = \{s_2, s_3\}$, and $s_{1,t-1}$ is dependent on $o_t$ given $\{\mathbf{y}_{1:t-1}, a_{1:t-1}, s_{2,t-1}, s_{3,t-1}\}$. Hence, conditioning on $\tilde{\vec{s}}_{t-1}$ and $\tilde{\vec{s}}_{t-1}^{\text{ASR}}$ give different conditional mutual information. Therefore, in the above conditional mutual information, we need to condition on the previous states $\tilde{\vec{s}}_{t-1}$.

Moreover, the conditional mutual information $I(\mathbf{y}_t; \tilde{\vec{s}}_t^{\text{ASR}} | \mathbf{y}_{1:t-1}, a_{1:t-1}, \tilde{\vec{s}}_{t-1})$ can be upper bound by a KL-divergence:

$$
\begin{aligned}
& I(\mathbf{y}_t; \tilde{\vec{s}}_t^{\text{ASR}} | \mathbf{y}_{1:t-1}, a_{1:t-1}, \tilde{\vec{s}}_{t-1}) \\
& = I(\mathbf{y}_t; \tilde{\vec{s}}_t^{\text{ASR}}, \{\mathbf{y}_{1:t-1}, a_{1:t-1}, \tilde{\vec{s}}_{t-1}\}) - I(\mathbf{y}_t; \{\mathbf{y}_{1:t-1}, a_{1:t-1}, \tilde{\vec{s}}_{t-1}\}) \\
& = \left[ H(\mathbf{y}_t) - H(\mathbf{y}_t | \tilde{\vec{s}}_t^{\text{ASR}}, \mathbf{y}_{1:t-1}, a_{1:t-1}, \tilde{\vec{s}}_{t-1}) \right] - \left[ H(\mathbf{y}_t) - H(\mathbf{y}_t | \mathbf{y}_{1:t-1}, a_{1:t-1}, \tilde{\vec{s}}_{t-1}) \right] \\
& = H(\mathbf{y}_t | \mathbf{y}_{1:t-1}, a_{1:t-1}, \tilde{\vec{s}}_{t-1}) - H(\mathbf{y}_t | \tilde{\vec{s}}_t^{\text{ASR}}, \mathbf{y}_{1:t-1}, a_{1:t-1}, \tilde{\vec{s}}_{t-1}) \\
& = -p(\mathbf{y}_t | \mathbf{y}_{1:t-1}, a_{1:t-1}, \tilde{\vec{s}}_{t-1}) \log p(\mathbf{y}_t | \mathbf{y}_{1:t-1}, a_{1:t-1}, \tilde{\vec{s}}_{t-1}) \\
& \quad + p(\mathbf{y}_t | \tilde{\vec{s}}_t^{\text{ASR}}, \mathbf{y}_{1:t-1}, a_{1:t-1}, \tilde{\vec{s}}_{t-1}) \log p(\mathbf{y}_t | \tilde{\vec{s}}_t^{\text{ASR}}, \mathbf{y}_{1:t-1}, a_{1:t-1}, \tilde{\vec{s}}_{t-1}) \\
& \leq -p(\mathbf{y}_t | \tilde{\vec{s}}_t^{\text{ASR}}, \mathbf{y}_{1:t-1}, a_{1:t-1}, \tilde{\vec{s}}_{t-1}) \log p(\mathbf{y}_t | \mathbf{y}_{1:t-1}, a_{1:t-1}, \tilde{\vec{s}}_{t-1}) \\
& \quad + p(\mathbf{y}_t | \tilde{\vec{s}}_t^{\text{ASR}}, \mathbf{y}_{1:t-1}, a_{1:t-1}, \tilde{\vec{s}}_{t-1}) \log p(\mathbf{y}_t | \tilde{\vec{s}}_t^{\text{ASR}}, \mathbf{y}_{1:t-1}, a_{1:t-1}, \tilde{\vec{s}}_{t-1}) \\
& = \int q_\phi(\tilde{\vec{s}}_t^{\text{ASR}} | \tilde{\vec{s}}_{t-1}, \mathbf{y}_{1:t}, a_{1:t-1}) \log \frac{q_\phi(\tilde{\vec{s}}_t^{\text{ASR}} | \tilde{\vec{s}}_{t-1}, \mathbf{y}_{1:t}, a_{1:t-1})}{p_\gamma(\tilde{\vec{s}}_t^{\text{ASR}} | \tilde{\vec{s}}_{t-1}, a_{t-1}; D_{\vec{s}}, D_{a \to \vec{s}})} \, d\tilde{\vec{s}}_t^{\text{ASR}} \\
& = \text{KL}\big(q_\phi(\tilde{\vec{s}}_t^{\text{ASR}} | \tilde{\vec{s}}_{t-1}, \mathbf{y}_{1:t}, a_{1:t-1}) \| p_\gamma(\tilde{\vec{s}}_t^{\text{ASR}} | \tilde{\vec{s}}_{t-1}, a_{t-1}; D_{\vec{s}}, D_{a \to \vec{s}})\big),
\end{aligned}
$$

with $p_\gamma$ being the transition dynamics of $\tilde{\vec{s}}_t$ with parameters $\gamma$.

## C  IDENTIFIABILITY IN LINEAR-GAUSSIAN CASES

Below, we first show the identifiability guarantee in the linear case, as a special case of Eq. 2. In the linear case (see the environment model given in Eq. 5 in Appendix D), $D_{\vec{s} \to o}$, $D_{\vec{s} \to r}$, $D_{a \to r}$, $D_{\vec{s}}$, and $D_{a \to \vec{s}}$ are linear coefficients, indicating corresponding graph structures and also the strength. Denote the covariance matrices of $e_t$ and $\epsilon_t$ by $\Sigma_e$ and $\Sigma_\epsilon$, respectively. Further let $\ddot{D}_{\vec{s} \to o} := (D_{\vec{s} \to o}^\top, D_{\vec{s} \to r}^\top)^\top$. The following proposition shows that the environment model in the linear case is identifiable up to some orthogonal transformation on certain coefficient matrices from observed data $\{\langle o_t, a_t, r_t \rangle\}_{t=1}^T$.

**Proposition 2** (Identifiability). *Suppose the perceived signal $o_t$, the reward $r_t$, and the latent states $\vec{s}_t$ follow a linear environment model. If assumptions A1~A4 (given in Appendix C.1) hold and with the second-order statistics of the observed data $\{\langle o_t, a_t, r_t \rangle\}_{t=1}^T$, the noise variances $\Sigma_e$ and $\Sigma_\epsilon$, $D_{a \to r}$, $\ddot{D}_{\vec{s} \to o} D_{\vec{s}}^k D_{a \to \vec{s}}$ (with $k \geq 0$), and $\ddot{D}_{\vec{s} \to o} \ddot{D}_{\vec{s} \to o}^\top$ are uniquely identified.*

This proposition shows that in the linear case, with the second-order statistics of the observed data, we can identify the parameters up to orthogonal transformations. In particular, suppose the linear environment model with parameters $(D_{\vec{s} \to o}, D_{\vec{s} \to r}, D_{a \to r}, D_{\vec{s}}, D_{a \to \vec{s}}, \Sigma_e, \Sigma_\epsilon)$ and that with $(\tilde{D}_{\vec{s} \to o}, \tilde{D}_{\vec{s} \to r}, \tilde{D}_{a \to r}, \tilde{D}_{\vec{s}}, \tilde{D}_{a \to \vec{s}}, \tilde{\Sigma}_{\tilde{e}}, \tilde{\Sigma}_{\tilde{\epsilon}})$ are observationally equivalent. Then we have $\tilde{D}_{\vec{s} \to o} = \ddot{D}_{\vec{s} \to o} U$, $\tilde{D}_{a \to r} = D_{a \to r}$, $\tilde{D}_{\vec{s}} = U^\top D_{\vec{s}} U$, $\tilde{D}_{a \to \vec{s}} = D_{a \to \vec{s}} U$, $\tilde{\Sigma}_{\tilde{e}} = \Sigma_e$, and $\tilde{\Sigma}_{\tilde{\epsilon}} = \Sigma_\epsilon$, where $U$ is an orthogonal matrix.

### C.1  ASSUMPTIONS OF PROPOSITION 2

To show the identifiability of the model in the linear case, we make the following assumptions:

A1.  $d_o + d_r \geq d_s$, where $|o_t| = d_o$, $|r_t| = d_r$, and $|s_t| = d_s$.
A2.  $(D_{\vec{s} \to o}^\top, D_{\vec{s} \to r}^\top)^\top$ is full column rank and $D_{\vec{s}}$ is full rank.
A3.  The control signal $a_t$ is i.i.d. and the state $\vec{s}_t$ is stationary.
A4.  The process noise has a unit variance, i.e., $\text{var}(\eta_t) = I$.

### C.2  PROOF OF PROPOSITION 2

*Proof.* The proof of the linear case without control signals has been shown in Zhang & Hyvärinen (2011). Below, we give the identifiability proof in the linear-Gaussian case with control signals:

$$
\begin{cases}
o_t = D_{\vec{s} \to o} \vec{s}_t + e_t, \\
r_{t+1} = D_{\vec{s} \to r} \vec{s}_t + D_{a \to r} a_t + \epsilon_{t+1}, \\
\vec{s}_t = D_{\vec{s}} \vec{s}_{t-1} + D_{a \to \vec{s}} a_{t-1} + \eta_t.
\end{cases}
\tag{5}
$$

Let $\mathbf{y}_{t+1} = [o_t^\top, r_{t+1}^\top]^\top$, $\ddot{D}_{\vec{s} \to o} = [D_{\vec{s} \to o}^\top, D_{\vec{s} \to r}^\top]^\top$, $\ddot{D}_{a \to r} = [\vec{0}^\top, D_{a \to r}^\top]^\top$, and $\ddot{e}_t = [e_t^\top, \epsilon_{t+1}^\top]^\top$. Then the above equation can be represented as:

$$
\begin{cases}
\mathbf{y}_t = \ddot{D}_{\vec{s} \to o} \vec{s}_t + \ddot{D}_{a \to r} a_t + \ddot{e}_t, \\
\vec{s}_t = D_{\vec{s}} \vec{s}_{t-1} + D_{a \to \vec{s}} a_{t-1} + \eta_t.
\end{cases}
\tag{6}
$$

Because the dynamic system is linear and Gaussian, we make use of the second-order statistics of the observed data to show the identifiability. We first consider the cross-covariance between $\mathbf{y}_{t+k}$ and $a_t$:

$$\begin{cases} \mathrm{Cov}(\mathbf{y}_{t+k}, a_t) = \ddot{D}_{\vec{s} \to o} D_{\vec{s}}^{k-1} D_{a \to \vec{s}} \cdot \mathrm{Var}(a_t), & \text{if } k > 0. \\ \mathrm{Cov}(\mathbf{y}_{t+k}, a_t) = \ddot{D}_{a \to r} \cdot \mathrm{Var}(a_t), & \text{if } k = 0. \end{cases} \quad (7)$$

Thus, from the cross-covariance between $\mathbf{y}_{t+k}$ and $a_t$, we can identify $\ddot{D}_{\vec{s} \to o} D_{a \to \vec{s}}$, $\ddot{D}_{a \to r}$, and $\ddot{D}_{\vec{s} \to o} D_{\vec{s}}^{k} D_{a \to \vec{s}}$ for $k > 0$.

Next, we consider the auto-covariance function of $\vec{s}$. Define the auto-covariance function of $\vec{s}$ at lag $k$ as $\mathbf{R}_{\vec{s}}(k) = \mathbb{E}[\vec{s}_t \vec{s}_{t+k}^\top]$, and similarly for $\mathbf{R}_{\mathbf{y}}(k)$. Clearly, $\mathbf{R}_{\vec{s}}(-k) = \mathbf{R}_{\vec{s}}(k)^\top$ and $\mathbf{R}_{\mathbf{y}}(-k) = \mathbf{R}_{\mathbf{y}}(k)^\top$. Then we have

$$\begin{cases} \mathbf{R}_{\vec{s}}(k) = \mathbf{R}_{\vec{s}}(k-1) \cdot D_{\vec{s}}^\top, & \text{if } k > 0, \\ \mathbf{R}_{\vec{s}}(k) = \mathbf{R}_{\vec{s}}^\top(1) \cdot D_{\vec{s}}^\top + D_{a \to \vec{s}} \mathrm{Var}(a_{t-1}) D_{a \to \vec{s}}^\top + I, & \text{if } k = 0. \end{cases} \quad (8)$$

Below, we first consider the case where $d_o + d_r = d_s$. Let $\tilde{\mathbf{y}}_t = \ddot{D}_{\vec{s} \to o} \vec{s}_t$, so $\mathbf{y}_t = \tilde{\mathbf{y}}_t + \ddot{D}_{a \to r} a_{t-1} + \ddot{e}_t$ and $\mathbf{R}_{\tilde{\mathbf{y}}}(k) = \ddot{D}_{\vec{s} \to o} \mathbf{R}_{\vec{s}_t}(k) \ddot{D}_{\vec{s} \to o}^\top$. $\mathbf{R}_{\tilde{\mathbf{y}}}(k)$ satisfies the recursive property:

$$\begin{cases} \mathbf{R}_{\tilde{\mathbf{y}}}(k) = \mathbf{R}_{\tilde{\mathbf{y}}}(k-1) \cdot \Omega^\top, & \text{if } k > 0, \\ \mathbf{R}_{\tilde{\mathbf{y}}}(k) = \mathbf{R}_{\tilde{\mathbf{y}}}^\top(1) \cdot \Omega^\top + \ddot{D}_{\vec{s} \to o}(D_{a \to \vec{s}} \mathrm{Var}(a_{t-1}) D_{a \to \vec{s}}^\top + I) \ddot{D}_{\vec{s} \to o}^\top, & \text{if } k = 0, \end{cases} \quad (9)$$

where $\Omega = \ddot{D}_{\vec{s} \to o} D_{\vec{s}} \ddot{D}_{\vec{s} \to o}^{-1}$.

Denote $S_k = \ddot{D}_{\vec{s} \to o} D_{\vec{s}}^{k-1} D_{a \to \vec{s}} \cdot \mathrm{Var}(a_t)$. Then we can derive the recursive property for $\mathbf{R}_{\mathbf{y}}(k)$:

$$\begin{cases} \mathbf{R}_{\mathbf{y}}(k) = \mathbf{R}_{\mathbf{y}}(k-1) \cdot \Omega^\top - \ddot{D}_{a \to r} S_{k-1}^\top \Omega^\top + \ddot{D}_{a \to r} S_k^\top, & \text{if } k > 1, \\ \mathbf{R}_{\mathbf{y}}(k) = \mathbf{R}_{\mathbf{y}}(k-1) \cdot \Omega^\top - \ddot{D}_{a \to r} \mathrm{Var}^\top(a_t) \ddot{D}_{a \to r}^\top \Omega^\top - \Sigma_e \Omega^\top + \ddot{D}_{a \to r} S_k^\top, & \text{if } k = 1, \\ \mathbf{R}_{\mathbf{y}}(k) = \mathbf{R}_{\mathbf{y}}^\top(1) \cdot \Omega^\top + \left( \ddot{D}_{a \to r} \mathrm{Var}(a_t) \ddot{D}_{a \to r}^\top + \Sigma_e \right) \\ \qquad\qquad + \ddot{D}_{\vec{s} \to o}(D_{a \to \vec{s}} \mathrm{Var}(a_t) D_{a \to \vec{s}}^\top + I) \ddot{D}_{\vec{s} \to o}^\top, & \text{if } k = 0. \end{cases}$$

When $k = 2$, we have

$$\mathbf{R}_{\mathbf{y}}(2) = \mathbf{R}_{\mathbf{y}}(1) \cdot \Omega^\top - \ddot{D}_{a \to r} S_1^\top \Omega^\top + \ddot{D}_{a \to r} S_2^\top.$$

The above equation can be re-organized as

$$\left( \mathbf{R}_{\mathbf{y}}(2) - \ddot{D}_{a \to r} \cdot S_2^\top \right) = \left( \mathbf{R}_{\mathbf{y}}(1) - \ddot{D}_{a \to r} \cdot S_1^\top \right) \cdot \Omega^\top.$$

Because $\ddot{D}_{a \to r}$ and $S_k$ are identifiable, and suppose $\left( \mathbf{R}_{\mathbf{y}}(1) - \ddot{D}_{a \to r} \cdot S_1^\top \right)$ is invertible, $\Omega = \ddot{D}_{\vec{s} \to o} D_{\vec{s}} \ddot{D}_{\vec{s} \to o}^{-1}$ is identifiable.

We further consider $\mathbf{R}_{\mathbf{y}}(0)$ and $\mathbf{R}_{\mathbf{y}}(1)$ and write down the following form:

$$\begin{bmatrix} \mathbf{R}_{\mathbf{y}}(0) - \ddot{D}_{\vec{s} \to o}(D_{a \to \vec{s}} \mathrm{Var}(a_{t-1}) D_{a \to \vec{s}}^\top + I) \ddot{D}_{\vec{s} \to o}^\top \\ \mathbf{R}_{\mathbf{y}}(1) \end{bmatrix}$$

$$= \begin{bmatrix} \mathbf{R}_{\mathbf{y}}^\top(1) \\ \mathbf{R}_{\mathbf{y}}(0) \end{bmatrix} \cdot \Omega^\top + \begin{bmatrix} \ddot{D}_{a \to r} \mathrm{Var}(a_t) \ddot{D}_{a \to r}^\top \\ -\ddot{D}_{a \to r} \mathrm{Var}^\top(a_t) \ddot{D}_{a \to r}^\top \Omega^\top + \ddot{D}_{a \to r} S_1^\top \end{bmatrix} + \Sigma_e \begin{bmatrix} I \\ -\Omega^\top \end{bmatrix}.$$

From the above two equations we can then identify $\Sigma_e$ and $\ddot{D}_{\vec{s} \to o}(D_{a \to \vec{s}} \mathrm{Var}(a_{t-1}) D_{a \to \vec{s}}^\top + I) \ddot{D}_{\vec{s} \to o}^\top$, and because $\ddot{D}_{\vec{s} \to o} D_{a \to \vec{s}}$ is identifiable, $\ddot{D}_{\vec{s} \to o} \ddot{D}_{\vec{s} \to o}^\top$ is identifiable.

In summary, we have shown the identifiability of $\ddot{D}_{a \to r}$, $\ddot{D}_{\vec{s} \to o} D_{a \to \vec{s}}$, $\ddot{D}_{\vec{s} \to o} D_{\vec{s}}^k D_{a \to \vec{s}}$, $\ddot{D}_{\vec{s} \to o} \ddot{D}_{\vec{s} \to o}^\top$, and $\Sigma_e$. Furthermore, $\ddot{D}_{\vec{s} \to o}$, $D_{\vec{s}}$, and $D_{a \to \vec{s}}$ are identified up to some orthogonal transformations. That is, suppose the model in Eq. (3) with parameters $(D_{\vec{s} \to o}, D_{\vec{s} \to r}, D_{a \to r}, D_{\vec{s}}, D_{a \to \vec{s}}, \Sigma_e, \Sigma_\epsilon)$ and that with $(\tilde{D}_{\vec{s} \to o}, \tilde{D}_{\vec{s} \to r}, \tilde{D}_{a \to r}, \tilde{D}_{\vec{s}}, \tilde{D}_{a \to \vec{s}}, \tilde{\Sigma}_{\tilde{e}}, \tilde{\Sigma}_{\tilde{\epsilon}})$ are observationally equivalent, we then have $\ddot{D}_{\vec{s} \to o} = \ddot{D}_{\vec{s} \to o} U$, $\tilde{D}_{a \to r} = D_{a \to r}$, $\tilde{D}_{\vec{s}} = U^\top D_{\vec{s}} U$, $\tilde{D}_{a \to \vec{s}} = D_{a \to \vec{s}} U$, $\tilde{\Sigma}_{\tilde{e}} = \Sigma_e$, and $\tilde{\Sigma}_{\tilde{\epsilon}} = \Sigma_\epsilon$, where $U$ is an orthogonal matrix.

Next, we extend the above results to the case where $d_o + d_r > d_s$. Let $\ddot{D}_{\vec{s} \to o(i,\cdot)}^\top$ be the $i$-th row of $\ddot{D}_{\vec{s} \to o}$. Recall that $\ddot{D}_{\vec{s} \to o}$ is of full column rank. Then for any $i$, one can show that there always exist $d_s - 1$ rows

of $\ddot{D}_{\vec{s} \nrightarrow o}$, such that they, together with $\ddot{D}_{\vec{s} \nrightarrow o(i,\cdot)}^{\top}$, form a $d_s \times d_s$ full-rank matrix, denoted by $\ddot{\bar{D}}_{\vec{s} \nrightarrow o(i,\cdot)}$. Then from the observed data corresponding to $\ddot{\bar{D}}_{\vec{s} \nrightarrow o(i,\cdot)}$, $\ddot{\bar{D}}_{\vec{s} \nrightarrow o(i,\cdot)}$ is determined up to orthogonal transformations. Thus, $\ddot{D}_{\vec{s} \nrightarrow o}$ is identified up to orthogonal transformations. Similarly, $D_{a \nrightarrow r}$, $D_{\vec{s}}$, and $D_{a \nrightarrow \vec{s}}$ are identified up to orthogonal transformations. Furthermore, $\text{Cov}(\ddot{D}_{\vec{s} \nrightarrow o}\vec{s}_t + D_{a \nrightarrow r}a_t)$ is determined by $\ddot{D}_{\vec{s} \nrightarrow o}$, $\ddot{D}_{a \nrightarrow r}$, $D_{\vec{s}}$, and $D_{a \nrightarrow \vec{s}}$. Because $\text{Cov}(\mathbf{y}_t) = \text{Cov}(\ddot{D}_{\vec{s} \nrightarrow o}\vec{s}_t + D_{a \nrightarrow r}a_t) + \Sigma_{\ddot{e}}$, $\Sigma_{\ddot{e}}$ is identifiable.

One may further add sparsity constraints on $D_{\vec{s} \nrightarrow o}$, $D_{\vec{s} \nrightarrow r}$, $D_{\vec{s}}$, and $D_{a \nrightarrow \vec{s}}$, to select more sparse structures among the equivalent ones. For example, one may add sparsity constraints on the columns of $D_{\vec{s} \nrightarrow o}$. Note this corresponds to the mask on the elements of $\vec{s}_t$ in Eq. 2; if the full column is 0, then the corresponding dimension of $\vec{s}_t$ is not selected.

$\square$

## D   MORE EXPLANATIONS ON THE OBJECTIVE FUNCTION FOR GENERAL NONLINEAR MODELS

Below are the details of each component in the above objective function:

- Reconstruction and prediction components: These two parts are commonly used in sequential VAE. They aim to minimize the reconstruction error and prediction error of the perceived signal $o_t$ and the reward $r_t$.

- Transition component: To achieve the property that state representations satisfy an MDP, we explicitly model the transition dynamics: $\log p_\gamma(\tilde{\vec{s}}_t | \tilde{\vec{s}}_{t-1}, a_{t-1}; D_{\vec{s}}, D_{a \nrightarrow \vec{s}})$. In particular, $\tilde{\vec{s}}_t | \tilde{\vec{s}}_{t-1}$ is modelled with a mixture of Gaussians: $\sum_{k=1}^{K} \pi_k \mathcal{N}\big(\boldsymbol{\mu}_k(\tilde{\vec{s}}_{t-1}, a_{t-1}), \Sigma_k(\tilde{\vec{s}}_{t-1}, a_{t-1})\big)$, where $K$ is the number of mixtures, $\boldsymbol{\mu}_k(\cdot)$ and $\Sigma_k(\cdot)$ are given by multi-layer perceptrons (MLP) with inputs $\tilde{\vec{s}}_{t-1}$ and $a_{t-1}$, parameters $\gamma$, and structural constraints $D_{\vec{s}}$ and $D_{a \nrightarrow \vec{s}}$. This explicit constraint on state dynamics is essential for establishing a Markov chain in latent space and for learning a representation for long-term predictions. Note that unlike in traditional VAE (Kingma & Welling, 2013), we do not assume that different dimensions in $\tilde{\vec{s}}_t$ are marginally independent, but model their structural relationships explicitly to achieve conditional independence.

- KL-divergence constraint: The KL divergence is used to constrain the state space with multiple purposes: (1) It is used in the lower bound of $\log P(\mathbf{y}_{1:T})$ to achieve conditional disentanglement between $q_\phi(\tilde{s}_{i,t}|\cdot)$ and $q_\phi(\tilde{s}_{j,t}|\cdot)$ for $i \neq j$, (2) and also to achieve minimality of ASRs.

- Sufficiency & minimality constraints: We achieve minimal sufficient state representations for the downstream policy learning by leveraging action prediction given the cumulative reward, the conditional mutual information between $\mathbf{y}_t$ and $\tilde{\vec{s}}_t^{\text{ASR}}$, and structural constraints. For details, please refer to Section 2.1.

- Sparsity constraints: According to the edge-minimality property (Zhang & Spirtes, 2011), we additionally put sparsity constraints on structural matrices to achieve better identifiability. In particular, we use $L_1$ norm of the structural matrices as regularizers in the objective function to achieve sparsity of the solution.

Fig. 3 gives the diagram of the neural network architecture in model training. We use SS-VAE to learn the environment model and ASRs. Specifically, the encoder, which is used to learn the inference model $q_\phi(\tilde{\vec{s}}_t | \tilde{\vec{s}}_{t-1}, \mathbf{y}_{1:t}, a_{1:t-1})$, includes a Long Short-Term Memory (LSTM (Hochreiter & Schmidhuber, 1997)) to encode the sequential information with output $h_t$ and a Mixture Density Network (MDN (Bishop, 1994)) to output the parameters of MoGs. At each time instance, the input $\langle o_{t+1}, r_{t+1}, a_t \rangle$ is projected to the encoder and a sample of $\tilde{\vec{s}}_{t+1}$ is inferred from $q_\phi$ as output. The generated sample further acts as an input to the decoder, together with $a_{t+1}$ and structural matrices $D_{\vec{s} \nrightarrow o}$, $D_{\vec{s} \nrightarrow r}$, and $D_{a \nrightarrow r}$. Then the decoder outputs $\hat{o}_{t+1}$ and $\hat{r}_{t+2}$. Moreover, the state dynamics which satisfies a Markov process and is embedded with structural constraints $D_{\vec{s}}$ and $D_{a \nrightarrow \vec{s}}$, is modeled with an MLP and MDN, marked with red in Fig. 3. The action prediction part (denoted by *AP*), which helps sufficient state representations, uses MLP and is marked with blue. During training, we approximate the expectation in $\mathcal{L}$ by sampling and then jointly learn all parameters by maximizing $\mathcal{L}$ using stochastic gradient descent.

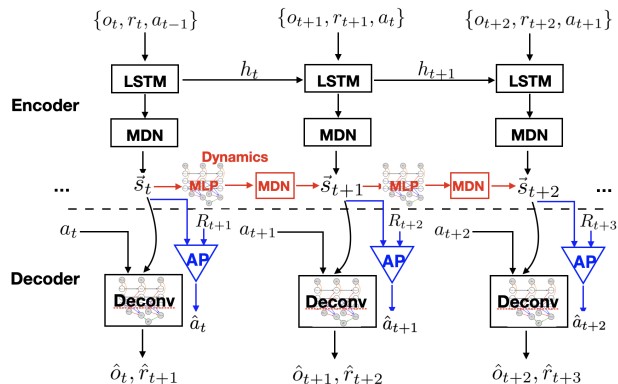

Figure 3: Diagram of neural network architecture to learn state representations. The corresponding structural constraints are involved in "Deconv" and "MLP", and "AP" represents the action prediction part for sufficient state representation learning.

## E MORE ESTIMATION DETAILS FOR GENERAL NONLINEAR MODELS

The generative model $p_\theta$ can be further factorized as follows:

$$
\begin{aligned}
&\log p_\theta(\mathbf{y}_{1:T}|\tilde{\vec{s}}_{1:T}, a_{1:T-1}; D_{\vec{s}\to o}, D_{\vec{s}\to r}, D_{a\to r}) \\
=\ &\log p_\theta(o_{1:T}|\tilde{\vec{s}}_{1:T}; D_{\vec{s}\to o}) + \log p_\theta(r_{1:T}|\tilde{\vec{s}}_{1:T}, a_{1:T-1}; D_{\vec{s}\to r}, D_{a\to r}) \\
=\ &\textstyle\sum_{t=1}^{T} \log p_\theta(o_t|\tilde{\vec{s}}_t; D_{\vec{s}\to o}) + \log p_\theta(r_t|\tilde{\vec{s}}_{t-1}, a_{t-1}; D_{\vec{s}\to r}, D_{a\to r}),
\end{aligned}
\tag{10}
$$

where both $p_\theta(o_t|\tilde{\vec{s}}_t; D_{\vec{s}\to o})$ and $p_\theta(r_t|\tilde{\vec{s}}_{t-1}, a_{t-1}; D_{\vec{s}\to r}, D_{a\to r})$ are modelled by mixture of Gaussians, with $D_{\vec{s}\to o}$ indicating the existence of edges from $\tilde{\vec{s}}_t$ to $o_t$ and $D_{\vec{s}\to r}$ indicating the existence of edges from $\tilde{\vec{s}}_{t-1}$ to $r_t$.

The inference model $q_\phi(\tilde{\vec{s}}_{1:T}|\mathbf{y}_{1:T}, a_{1:T-1})$ is factorized as

$$
\begin{aligned}
&\log q_\phi(\tilde{\vec{s}}_{1:T}|\mathbf{y}_{1:T}, a_{1:T-1}) \\
=\ &\log q_\phi(\tilde{\vec{s}}_1|\mathbf{y}_1, a_0) + \sum_{t=2}^{T} \log q_\phi(\tilde{\vec{s}}_t|\tilde{\vec{s}}_{t-1}, \mathbf{y}_{1:t}, a_{1:t-1}),
\end{aligned}
$$

where both $q_\phi(\tilde{\vec{s}}_1|\mathbf{y}_1, a_0)$ and $q_\phi(\tilde{\vec{s}}_t|\tilde{\vec{s}}_{t-1}, \mathbf{y}_{1:t}, a_{1:t-1})$ are modelled with mixture of Gaussians.

The transition dynamics $p_\gamma$ is factorized as

$$
\log p_\gamma(\tilde{\vec{s}}_{1:T}|a_{1:T-1}; D_{\vec{s}(\cdot,i)}, D_{a\to\vec{s}(\cdot,i)}) = \sum_{t=1}^{T} \log p_\gamma(\tilde{\vec{s}}_t|\tilde{\vec{s}}_{t-1}, a_{t-1}; D_{\vec{s}(\cdot,i)}, D_{a\to\vec{s}(\cdot,i)}),
\tag{11}
$$

with $\tilde{\vec{s}}_t|\tilde{\vec{s}}_{t-1}$ modelled with mixture of Gaussians.

Thus, the KL divergence can be represented as follows:

$$
\begin{aligned}
&\mathrm{KL}\big(q_\phi(\tilde{\vec{s}}_{1:T}|\mathbf{y}_{1:T}, a_{1:T-1})\|p_\gamma(\tilde{\vec{s}}_{1:T})\big) \\
=\ &\mathrm{KL}\big(q_\phi(\tilde{\vec{s}}_1|\mathbf{y}_1, a_0)\|p_\gamma(\tilde{\vec{s}}_1)\big) + \sum_{t=2}^{T} \mathbb{E}_{q_\phi}\big[\mathrm{KL}\big(q_\phi(\tilde{\vec{s}}_t|\tilde{\vec{s}}_{t-1}, \mathbf{y}_{1:t}, a_{1:t-1})\|p_\gamma(\tilde{\vec{s}}_t|\tilde{\vec{s}}_{t-1})\big)\big].
\end{aligned}
\tag{12}
$$

In practice, KL divergence with mixture of Gaussians is hard to implement, so instead, we used the following objective function:

$$
\begin{aligned}
&\mathrm{KL}\big(q_\phi(\tilde{\vec{s}}_1|\mathbf{y}_1, a_0)\|p_{\gamma'}(\tilde{\vec{s}}_1)\big) + \sum_{t=2}^{T} \mathbb{E}_{q_\phi}\big[\mathrm{KL}\big(q_\phi(\tilde{\vec{s}}_t|\tilde{\vec{s}}_{t-1}, \mathbf{y}_{1:t}, a_{1:t-1})\|p_{\gamma'}(\tilde{\vec{s}}_t|\tilde{\vec{s}}_{t-1})\big)\big] \\
&+\lambda \sum_{t=1}^{T} \log p_\gamma(\tilde{\vec{s}}_t|\tilde{\vec{s}}_{t-1}, a_{t-1}; D_{\vec{s}(\cdot,i)}, D_{a\to\vec{s}(\cdot,i)})
\end{aligned}
\tag{13}
$$

where $p_{\gamma'}$ is a standard multivariate Gaussian $\mathcal{N}(\vec{0}, I_d)$.

## F MORE DETAILS FOR POLICY LEARNING WITH ASRs

### F.1 MODEL-FREE AND MODEL-BASED POLICY LEARNING

**Model-Free Policy Learning.** For model-free policy learning, we make use of the learned environment model to infer ASRs $\tilde{\vec{s}}_t^{\mathrm{ASR}}$ from past observed sequences $\{o_{\leq t}, r_{\leq t}, a_{\leq t-1}\}$ and then

predict the action with the estimated low-dimensional ASRs. Our method is flexible to use a wide range of model-free methods; for example, one may use deep Q-learning for discrete actions (Mnih et al., 2015) and deep deterministic policy gradient (DDPG) for continuous actions (Lillicrap et al., 2015). Algorithm 1 gives the detailed procedure of model-free policy learning with ASRs in partially observable environments.

**Model-Based Policy Learning.** The downside of model-free RL algorithms is that they are usually data hungry, requiring very large amounts of interactions. On the contrary, model-based RL algorithms enjoy much better sample efficiency. Hence, we make use of the learned generative environment model, including the transition dynamics, observation function, and reward function, for model-based policy optimization. Based on the generative environment model, one can learn behaviors from imagined outcomes to increase sample-efficiency and mitigate heavy and possibly risky interactions with the environment. We present the procedure of the classic Dyna algorithm (Sutton, 1990; Sutton & Barto, 2018) with ASRs in Algorithm 2.

F.2    ALGORITHMS FOR POLICY LEARNING

Algorithm 1 gives the procedure of model-free policy learning with ASRs in partially observable environments. Specifically, it starts from model initialization (line 1) and data collection with a random policy (line 2). Then it updates the environment model and identifies the set of ASRs with the collected data (line 3), after which, the main procedure of policy optimization follows. In particular, because we do not directly observe the states $\vec{s}_t$, on lines 8 and 12, we infer $q_\phi(\vec{s}_{t+1}^{\text{ASR}}|o_{\leq t+1}, r_{\leq t+1}, a_{\leq t})$ and sample $\vec{s}_{t+1}^{\text{ASR}}$ from the posterior. The sampled ASRs are then stored in the buffer (line 13). Furthermore, we randomly sample a minibatch of $N$ transitions to optimize the policy (lines 14 and 15). One may perform various RL algorithms on the ASRs, such as deep deterministic policy gradient (DDPG (Lillicrap et al., 2015)) or Q-learning (Mnih et al., 2015).

Algorithm 2 presents the procedure of the classic Dyna algorithm with ASRs. Lines 17-22 make use of the learned environment model to predict the next step, including $\vec{s}_{t+1}^{\text{ASR}}$ and $r_{t+1}$, and update the Q function $n$ times. Specifically, in our implementation, the hyper-parameter $n$ is 20. Based on the learned model, the agent learns behaviors from imagined outcomes in the compact latent space, which helps to increase sample efficiency.

---

**Algorithm 1** Model-Free Policy Learning with ASRs in Partially Observable Environments

---

1: Randomly initialize neural networks and initialize replay buffer $\mathcal{B}$.
2: Apply random control signals and record multiple rollouts.
3: Estimate the model given in Eq. 2 with the recorded data (according to Section 3).
4: Identify indices of ASRs according to the learned graph structure and the criteria in Prop. 1.
5: **for** episode = 1, . . . , M **do**
6:     Initialize a random process $\mathcal{N}$ for action exploration.
7:     Receive initial observations $o_1$ and $r_1$.
8:     Infer the posterior $q_\phi(\vec{s}_1^{\text{ASR}}|o_1, r_1)$ and sample $\vec{s}_1^{\text{ASR}}$.
9:     **for** t = 1, . . . , T **do**
10:         Select action $a_t = \pi(\vec{s}_t^{\text{ASR}}) + \mathcal{N}_t$ according to the current policy and exploration noise.
11:         Execute action $a_t$ and receive reward $r_{t+1}$ and observation $o_{t+1}$.
12:         Infer the posterior $q_\phi(\vec{s}_{t+1}^{\text{ASR}}|o_{\leq t+1}, r_{\leq t+1}, a_{\leq t})$ and sample $\vec{s}_{t+1}^{\text{ASR}}$.
13:         Store transition $(\vec{s}_t^{\text{ASR}}, a_t, r_{t+1}, \vec{s}_{t+1}^{\text{ASR}})$ in $\mathcal{B}$.
14:         Sample a random minibatch of $N$ transitions $(\vec{s}_i^{\text{ASR}}, a_i, r_{i+1}, \vec{s}_{i+1}^{\text{ASR}})$ from $\mathcal{B}$.
15:         Update network parameters using a specified RL algorithm (e.g., DQN or DDPG).
16:     **end for**
17: **end for**

---

G    MAIN EXPERIMENTAL RESULTS

To evaluate the proposed approach, we conducted experiments on both CarRacing (Klimov, 2016) and VizDoom (Kempka et al., 2016) environments, following the setup in the world model (Ha & Schmidhuber, 2018) for a fair comparison. It is known that CarRacing is very challenging—the recent world model (Ha & Schmidhuber, 2018) is the first known solution to achieve the score required to

---

**Algorithm 2** Model-Based Policy Learning with ASRs in Partially Observable Environments

---

1: Randomly initialize neural networks and initialize replay buffer $\mathcal{B}$.
2: Apply random control signals and record multiple rollouts.
3: Estimate the model given in Eq. 2 with the recorded data (according to Section 3).
4: Identify indices of ASRs according to the learned graph structure and the criteria in Prop. 1.
5: **for** episode = 1, ..., M **do**
6:    Initialize a random process $\mathcal{N}$ for action exploration.
7:    Receive initial observations $o_1$ and $r_1$.
8:    Infer the posterior $q_\phi(\vec{s}_1^{\text{ASR}}|o_1, r_1)$ and sample $\vec{s}_1^{\text{ASR}}$.
9:    **for** t = 1, ..., T **do**
10:       Select action $a_t = \pi(\vec{s}_t^{\text{ASR}}) + \mathcal{N}_t$ according to the current policy and exploration noise.
11:       Execute action $a_t$ and receive reward $r_{t+1}$ and observation $o_{t+1}$.
12:       Infer the posterior $q_\phi(\vec{s}_{t+1}^{\text{ASR}}|o_{\leq t+1}, r_{\leq t+1}, a_{\leq t})$ and sample $\vec{s}_{t+1}^{\text{ASR}}$.
13:       Store transition $(\vec{s}_t^{\text{ASR}}, \vec{s}_t, a_t, r_{t+1}, \vec{s}_{t+1}^{\text{ASR}}, \vec{s}_{t+1}, o_{t+1})$ in $\mathcal{B}$.
14:       Sample a random minibatch of $N$ transitions $(\vec{s}_i^{\text{ASR}}, a_i, r_{i+1}, \vec{s}_{i+1}^{\text{ASR}})$ from $\mathcal{B}$.
15:       Update network parameters using a specified RL algorithm (e.g., DQN or DDPG).
16:       Update the model given in Eq. 2 with the recorded data from $\mathcal{B}$ (according to Section 3).
17:       **for** p = 1, ..., n **do**
18:          Sample a random minibatch of pairs of $(\vec{s}_t, a_t)$ from $\mathcal{B}$.
19:          Predict $(\vec{s}_{t+1}^{\text{ASR}}, r_{t+1})$ according to the model given in Eq. 2.
20:          Update network parameters using a specified RL algorithm (e.g., DQN or DDPG).
21:       **end for**
22:    **end for**
23: **end for**

---

solve the task. Without stated otherwise, all results were averaged across five random seeds, with standard deviation shown in the shaded area.

### G.1 CARRACING EXPERIMENT

CarRacing is a continuous control task with three continuous actions: steering left/right, acceleration, and brake. Reward is $-0.1$ every frame and $+1000/N$ for every track tile visited, where $N$ is the total number of tiles in track. It is obvious that the CarRacing environment is partially observable: by just looking at the current frame, although we can tell the position of the car, we know neither its direction nor velocity that are essential for controlling the car. For a fair comparison, we followed the same setting as in Ha & Schmidhuber (2018). Specifically, we collected a dataset of $10k$ random rollouts of the environment, each consisting of 1000 time steps. The dimensionality of latent states $\tilde{\vec{s}}_t$ was set to $\tilde{d} = 32$, determined by hyperparameter tuning.

**Analysis of ASRs.** To demonstrate the structures over observed frames, latent states, actions, and rewards, we visualized the learned $D_{\vec{s} \to o}$, $D_{\vec{s} \to r}$, $D_{\vec{s}}$, and $D_{a \to \vec{s}}$, as shown in Fig. 7 in Appendix H. Intuitively, we can see that $D_{\vec{s} \to r}$ and $D_{a \to \vec{s}}$ have many values close to zero, meaning that the reward is only influenced by a small number of state dimensions, and not many state dimensions are influenced by the action. Furthermore, from $D_{\vec{s}}$, we found that there are influences from $\tilde{\vec{s}}_{i,t}$ to $\tilde{\vec{s}}_{i,t+1}$ (diagonal values) for most state dimensions, which is reasonable because we want to learn an MDP over the underlying states, while the connections across states (off-diagonal values) are much sparser. Compared to the original 32-dim latent states, ASRs have only 21 dimensions. Below, we empirically showed that the low-dimensional ASRs significantly improve the policy learning performance in terms of both efficiency and efficacy.

**Comparison Between Model-Free and Model-Based ASRs.** We applied both model-free (DDPG) (Lillicrap et al., 2015) and model-based (Dyna and Prioritized Sweeping) algorithms (Sutton, 1990) to ASRs (with 21-dims). As shown in Fig. 2a, interestingly, by taking advantage of the learned generative model, model-based ASRs is superior to model-free ASRs at a faster rate, which demonstrates the effectiveness of the learned model. It also shows that with the estimated environment model and ASRs, we can learn behaviors from imagined outcomes to improve sample-efficiency.

**Comparison with VRL, SLAC, PlaNet, DBC, and Dreamer.** We also compared the proposed framework of policy learning with ASRs (with 21-dims) with 1) the same learning strategy but with vanilla representation learning (VRL, implemented without the components for minimal sufficient state representations as in Eq. 3), 2) SLAC (Lee et al., 2019), 3) PlaNet (Hafner et al., 2018), 4)

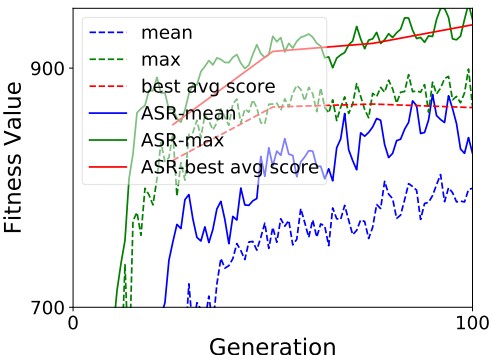

Figure 4: Fitness Value of ASRs compared to world models evaluated on CarRacing, including mean score, max score, and the best average score.

DBC (Zhang et al., 2021), and 5) Dreamer (Hafner et al., 2019). For a fair comparison, the latent dimensions of VRL, PlaNet, SLAC, DBC and Dreamer are set to 21 as well, and we require all of them to have the model capacity similar to ours (i.e., similar model architectures). From Fig. 2a, we can see that our methods, both model-free and model-based, obviously outperform others. It is worth noting that the huge performance difference between ASRs and VRL shows that the components for minimal sufficient state representations play a pivotal role in our objective.

**Comparison with World Models.** In light of the fact that world models (Ha & Schmidhuber, 2018) achieved good performance in CarRacing, we further compared our method (with 21-dim ASRs) with the world model. For a fair comparison, following Ha & Schmidhuber (2018), we also used the Covariance-Matrix Adaptation Evolution Strategy (CMA-ES) (Hansen, 2016) with a population of 64 agents to optimize the parameters of the controller. In addition, following the same setting as in Ha & Schmidhuber (2018) (where the agent's fitness value is defined as the average cumulative reward of the 16 random rollouts), we show the fitness values of the best performer (max) and the population (mean) at each generation (Fig. 4). We also took the best performing agent at the end of every 25 generations and tested it over 1024 random rollout scenarios to record the average (best avg score). It is obvious that our method (denoted by *ASR-\**) has a more efficient and also efficacy training process. The best average score of ASRs is 65 higher than that of world models.

**Comparison with Dreamer and DBC with Background Distraction.** We further compared ASRs (with 21-dims) with Dreamer and DBC when there are natural video distractors in CarRacing; we chose Dreamer and DBC, because their performance are relatively better than other comparisons when there are no distractors. Specifically, we followed Zhang et al. (2021) to incorporate natural video from the Kinetics dataset (Kay et al., 2017) as background in CarRacing. Similarly, for a fair comparison, we require all of them to have the same latent dimensions and have the similar model capacity. As shown in Fig. 2b, we can see that our method outperforms both Dreamer and DBC.

**Ablation Study.** We further performed ablation studies on latent dynamics prediction; that is, we compared with the case when the transition dynamics in Eq. 4 is not explicitly modeled, but is replaced with a standard normal distribution. Fig. 8 in Appendix H shows that by explicitly modelling the transition dynamics (denoted by *with LDP*), the cumulative reward has an obvious improvement over the one without modelling the transition dynamics (denoted by *without LDP*).

### G.2 VizDoom Experiment

We also applied the proposed method to VizDoom *take cover* scenario (Kempka et al., 2016), which is a discrete control problem with two actions: move left and move right. Reward is +1 at each time step while alive, and the cumulative reward is defined to be the number of time steps the agent manages to stay alive during an episode.

Considering that in the *take over* scenario the action space is discrete, we applied the widely used DQN (Mnih et al., 2013) on ASRs for policy learning. In addition to the comparisons with VRL (as in CarRacing) and DQN on raw observations, we further compared with another common approach to POMDPs: DRQN (Hausknecht & Stone, 2015). As shown in Fig. 2c, DQN on ASRs achieve a much better performance than all other comparisons, and in particular, DQN on ASRs outperforms DRQN on observations by around 400 on average in terms of cumulative reward. Similarly, we applied model-based (Dyna) algorithms (Sutton, 1990) to ASRs (with 21-dims). As shown in Fig. 2c, we can

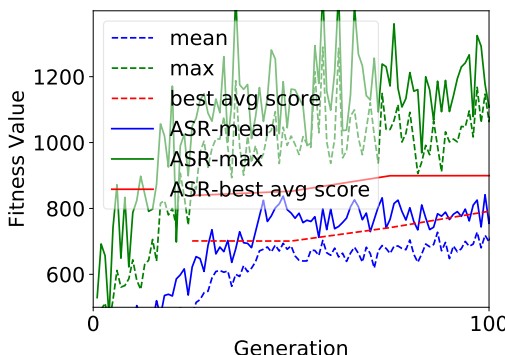

Figure 5: Fitness value of ASRs (with CMA-ES) compared to world models evaluated on VizDoom.

draw the same conclusion that by taking advantage of the learned generative model, model-based ASRs is superior to model-free ASRs at a faster rate. We also applied ASRs to world models, where Fig. 5 shows that our method with ASRs (denoted by *ASR-\**) achieves a better performance.

## H  ADDITIONAL EXPERIMENTS AND DETAILS

### H.1  CARRACING EXPERIMENT

CarRacing (with an illustration in Fig. 6) is a continuous control task with three continuous actions: steering left/right, acceleration, and brake. Reward is $-0.1$ every frame and $+1000/N$ for every track tile visited, where $N$ is the total number of tiles in track. It is obvious that the CarRacing environment is partially observable: by just looking at the current frame, although we can tell the position of the car, we know neither its direction nor velocity that are essential for controlling the car.

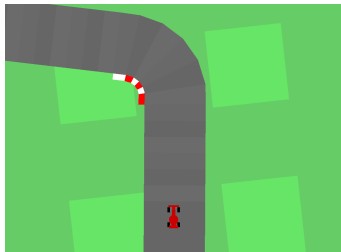

Figure 6: An illustration of Car Racing environment.

For a fair comparison, we followed the same setting as in Ha & Schmidhuber (2018). Specifically, we collected a dataset of $10k$ random rollouts of the environment, each consisting of $1000$ time steps, for model estimation. The dimensionality of latent states $\tilde{\vec{s}}_t$ was set to $\tilde{d} = 32$, and regularization parameters was set to $\lambda_1 = 1$, $\lambda_2 = 1$, $\lambda_3 = 1$, $\lambda_4 = 1$, $\lambda_5 = 1$, $\lambda_6 = 6$, $\lambda_7 = 10$, $\lambda_8 = 0.1$, which are determined by hyperparameter turning.

**Analysis of ASRs.**   To demonstrate the structures over observed frames, latent states, actions, and rewards, we visualized the learned $D_{\vec{s} \to o}$, $D_{\vec{s} \to r}$, $D_{\vec{s}}$, and $D_{a \to \vec{s}}$, as shown in Fig. 7. Intuitively, we can see that $D_{\vec{s} \to r}$ and $D_{a \to \vec{s}}$ have many values close to zero, meaning that the reward is only influenced by a small number of state dimensions, and not many state dimensions are influenced by the action. Furthermore, from $D_{\vec{s}}$, we found that there are influences from $\tilde{\vec{s}}_{i,t}$ to $\tilde{\vec{s}}_{i,t+1}$ (diagonal values) for most state dimensions, which is reasonable because we want to learn an MDP over the underlying states, while the connections across states (off-diagonal values) are much sparser. Compared to the original 32-dim latent states, ASRs have only 21 dimensions. Below, we empirically showed that the low-dimensional ASRs significantly improve the policy learning performance in terms of both efficiency and efficacy.

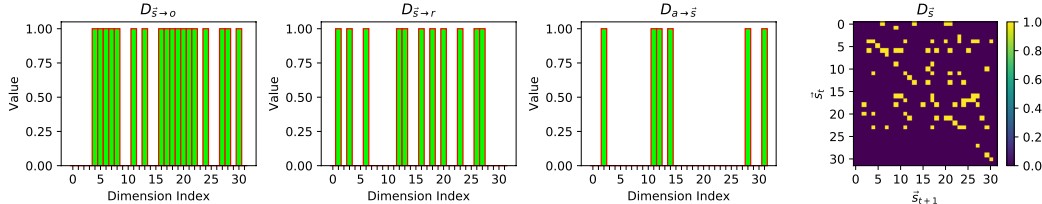

Figure 7: Visualization of estimated structural matrices $D_{\vec{s} \to o}$, $D_{\vec{s} \to r}$, $D_{a \to \vec{s}}$, and $D_{\vec{s}}$ in Car Racing.

**Ablation Study.** We further performed ablation studies on latent dynamics prediction; that is, we compared with the case when the transition dynamics in Fig. 4 are not explicitly involved. Fig. 8 shows that by explicitly modelling the transition dynamics (denoted by *with LDP*), the cumulative reward has an obvious improvement over the one without modelling the transition dynamics (denoted by *without LDP*).

**Difference between our SS-VAE and Planet, Dreamer.** Both our method and Planet (Hafner et al., 2018) and Dreamer (Hafner et al., 2019) are world model-based methods. The differences are mainly in two aspects: (1) our method explicitly considers the structural relationships among variables in the RL system, and (2) it guarantees minimal sufficient state representations for policy learning. Previous approaches usually fail to take into account whether the extracted state representations are sufficient and necessary for downstream policy learning. Moreover, as for the component of recurrent networks, SS-VAE uses LSTM that only contains the stochastic part, while PlaNet and Dreamer use RSSM that contains both deterministic and stochastic components.

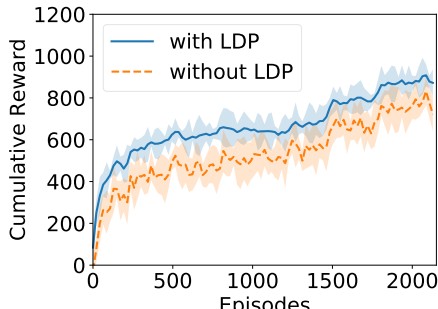

Figure 8: Ablation study of latent dynamics prediction (LDP) evaluated on Car Racing with model-free ASR.

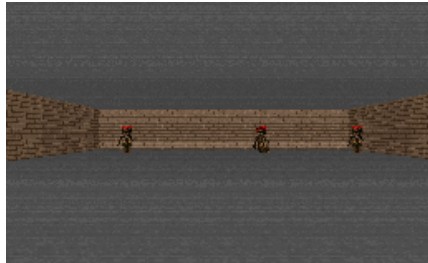

Figure 9: An illustration of VizDoom *take cover* scenario.

## H.2 VIZDOOM EXPERIMENT

We also applied the proposed method to VizDoom (Kempka et al., 2016). VizDoom provides many scenarios and we chose the *take cover* scenario (Fig. 9). Unlike CarRacing, *take cover* is a discrete control problem with two actions: move left and move right. Reward is +1 at each time step while alive, and the cumulative reward is defined to be the number of time steps the agent manages to stay alive during a episode. Therefore, in order to survive as long as possible, the agent has to learn how to avoid fireballs shot by monsters from the other side of the room. In this task, *solving* is defined as attaining the average survival time of greater than 750 time steps over 100 consecutive episodes, each running for a maximum of 2100 time steps.

Following the same setting as in Ha & Schmidhuber (2018), we collected a dataset of 10k random rollouts of the environment, each consisting of 500 time steps. The dimensionality of latent state $\tilde{\vec{s}}_t$ is set to $\tilde{d} = 32$. We also set $\lambda_1 = 1$, $\lambda_2 = 1$, $\lambda_3 = 1$, $\lambda_4 = 1$, $\lambda_5 = 1$, $\lambda_6 = 6$, $\lambda_7 = 10$, $\lambda_8 = 0.1$. By tuning thresholds, we finally reported all the results on the 21-dim ASRs, which achieved the best results in all the experiments.

# I  DETAILED MODEL ARCHITECTURES

In the car racing experiment, the original screen images were resized to $64 \times 64 \times 3$ pixels. The encoder consists of three components: a preprocessor, an LSTM, and an MDN. The preprocessor architecture is presented in Fig. 10, which takes as input the images, actions and rewards, and its output acts as the input to LSTM. We used 256 hidden units in the LSTM and used a five-component Gaussian mixture in the MDN. The decoder also consists of three components: a current observation reconstructor (Fig. 11), a next observation predictor (Fig. 12), and a reward predictor (Fig. 13). The architecture of the transition/dynamics is shown in Fig. 14, and its output is also modelled by an MDN with a five-component Gaussian mixture. The architecture of the action prediction is given in Fig. 15, which is a two-layer MLP taking states and rewards as input and predicted action as output. In the VizDoom experiment, we used the same image size and the same architectures except that the LSTM has 512 hidden units and the action has one dimension. It is worth emphasising that we applied weight normalization to all the parameters of the architectures above except for the structural matrices $D_{(\cdot)}$.

In DDPG, both actor network and critic network are modelled by two fully connected layers of size 300 with ReLU and batch normalisation. Similarly, in DQN (Mnih et al., 2013) on both ASRs and SSSs, the Q network is also modelled by two fully connected layers of size 300 with ReLU and batch normalisation. However, in DQN on observations, it is modelled by three convolutional layers (i.e., relu conv $32 \times 8 \times 8 \longrightarrow$ relu conv $64 \times 4 \times 4 \longrightarrow$ relu conv $64 \times 3 \times 3$) followed by two additional fully connected layers of size 64. In DRQN (Hausknecht & Stone, 2015) on observations, we used the same architecture as in DQN on observations but padded an extra LSTM layer with 256 hidden units as the final layer.

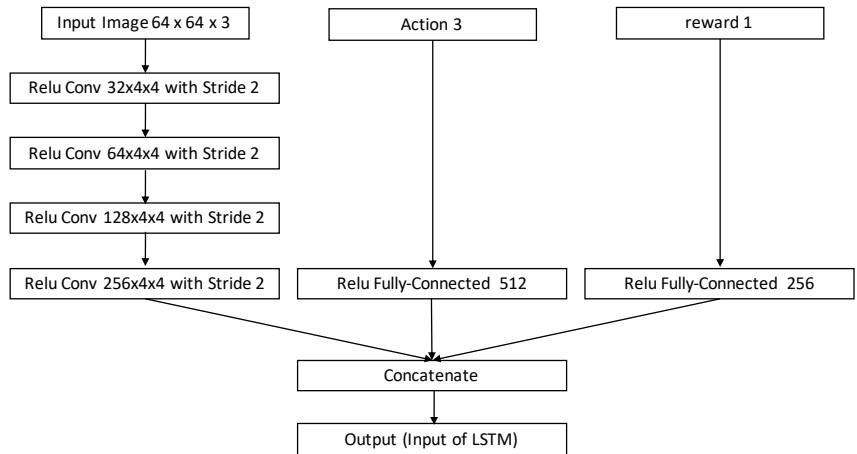

Figure 10: Network architecture of preprocessor.

# J  RELATED WORK

In the past few years, a number of approaches have been proposed to learn low-dimensional Markovian representations, which capture the variation in the environment generated by the agent's actions, without direct supervision (Lesort et al., 2018; Krishnan et al., 2015; Karl et al., 2016; Ha & Schmidhuber, 2018; Watter et al., 2015; Zhang et al., 2018; Kulkarni et al., 2016; Mahadevan & Maggioni, 2007; Gelada et al., 2019; Gregor et al., 2018; Ghosh et al., 2019; Zhang et al., 2021). Common strategies for such state representation learning include reconstructing the observation, learning a forward model, or learning an inverse model. Furthermore, prior knowledge, such as temporal continuity (Wiskott & Sejnowski, 2002), can be added to constrain the state space.

Recently, much attention has been paid to world models, which try to learn an abstract representation of both spatial and temporal aspects of the high-dimensional input sequences (Watter et al., 2015; Ebert et al., 2017; Ha & Schmidhuber, 2018; Hafner et al., 2018; Zhang et al., 2019b; Gelada et al., 2019; Kaiser et al., 2019; Hafner et al., 2019; 2020). Based on the learned world model, agents can perform model-based RL or planning. Our proposed method is also in the class of world models, which models the generative environment model, and additionally, encodes structural constraints

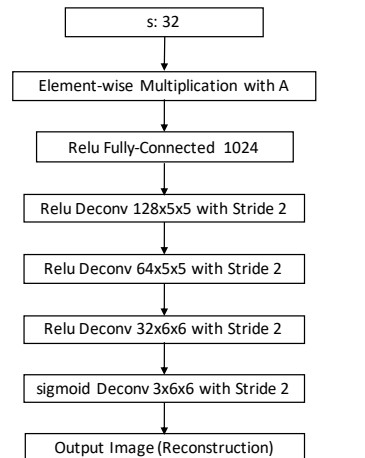

Figure 11: Network architecture of observation reconstruction.

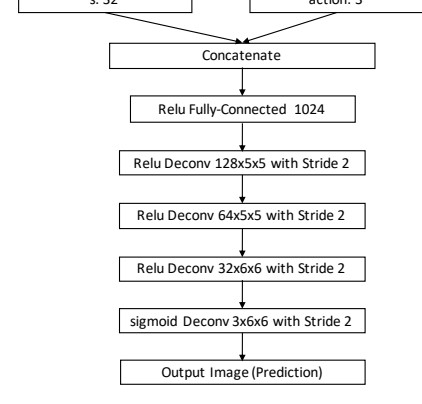

Figure 12: Network architecture of observation prediction.

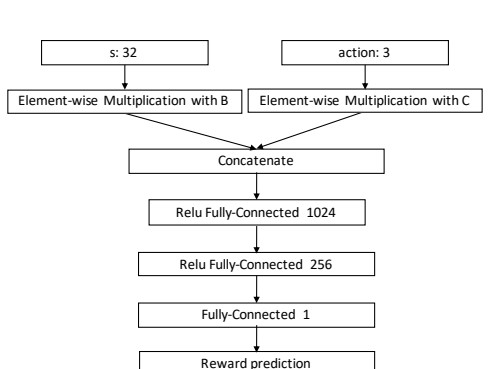

Figure 13: Network architecture of reward.

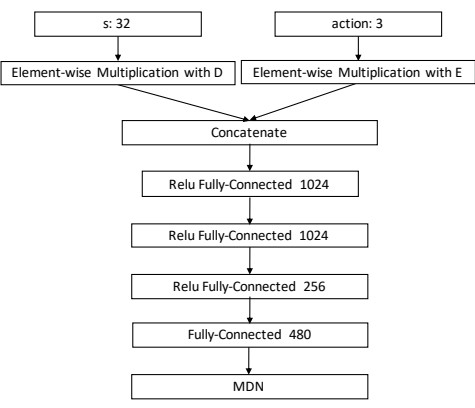

Figure 14: Network architecture of transition/dynamics.

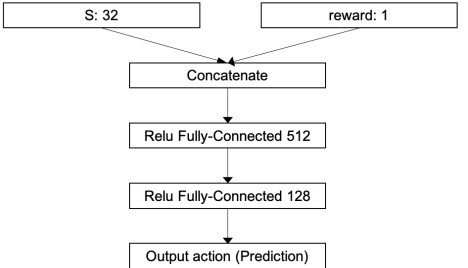

Figure 15: Network architecture of action prediction.



Figure 16: Visualization of estimated structural matrices $D_{\vec{s}\to o}$, $D_{\vec{s}\to r}$, $D_{a\to\vec{s}}$, and $D_{\vec{s}}$ in Car Racing, without the explicit sparsity constraints.



Figure 17: Visualization of estimated structural matrices $D_{\vec{s}\to o}$, $D_{\vec{s}\to r}$, $D_{a\to\vec{s}}$, and $D_{\vec{s}}$ in VizDoom.

and achieves the sufficiency and minimality of the estimated state representations from the view of generative and selection process. In contrast, Shu et al. (2020) makes use of contrastive loss, as an alternative of reconstruction loss; however, it only focuses on the transition dynamics and also fails to ensure the sufficiency and minimality. Another line of approaches of state representation learning is based on predictive state representations (PSRs) (Littman & Sutton, 2002; Singh et al., 2004). A recent approach generalizes PSRs to nonlinear predictive models, by exploiting the coarsest partition of histories into classes that are maximally predictive of the future (Zhang et al., 2019a). Moreover, bisimulation-based methods have also attracted much attention (Castro, 2020; Zhang et al., 2021).

On the other hand, our work is also related to Bayesian network learning and causal discovery (Spirtes et al., 1993; Pearl, 2000; Huang* et al., 2020). For example, Strehl et al. (2007) considers factorized-state MDP with structures being modeled with dynamic Bayesian network or decision trees. Incorporating such structure information has shown benefits in several machine learning tasks (Zhang* et al., 2020; Huang et al., 2019), and in this paper, we show its advantages in POMDPs.

## K  PLATFORM AND LICENSE

We run all the experiments on the servers with 4 NVidia V100 GPUs. We used the Car Racing in OpenAI gym and VizDoom environments and we have cited the creators. In our code, we have used the following libraries: Tensorflow (Apache License 2.0), OpenAI Gym (MIT License), VizDoom (MIT License), OpenCV (Apache 2 License), Numpy (BSD 3-Clause "New" or "Revised" License) and NVIDIA-DALI libraries (Apache 2 License).

