# OpenReview forum: "Action-Sufficient State Representation Learning for Control with Structural Constraints"
_ICLR.cc/2022/Workshop/OSC — ICLR2022 OSC  Poster_

### Official Review · Reviewer_6kLk · 2022-03-14
**Paper with strong technical depth**

**Rating:** 3
**Confidence:** 2

**Review:**

The authors introduce a novel generative RL environment graphical model. They introduce the concept of *Action-Sufficient state Representations* (ASRs), state representations that are minimally suffiicient to choose optimal actions, characterized conditions on the proposed environment model. The authors propose a SS-VAE model for explicitly learning ASRs, and achieve better results on the CarRacing and VizDoom environments against prior world-model related works (VRL, SLAC, PlaNet, DBC, Dreamer).

Strengths:
* Strong technical depth and detail, discussing derivation and relation to prior work
* Strong results on CarRacing and VizDoom

Weaknesses:
* Technical parts are difficult to follow
* Learning objective seems overly complex (8 parameters in equation 4)

Misc. Comments
* Baselines SLAC, PlaNet, Dreamer etc. all evaluate on the DeepMind Control Suite, why was this not evaluated here?
* Prior work (Predictive Information) [Lee, et al. 2020](https://proceedings.neurips.cc/paper/2020/hash/89b9e0a6f6d1505fe13dea0f18a2dcfa-Abstract.html) has also tackled learning minimally sufficient state representations, but with a contrastive learning objective similar to CURL. I am wondering how the state representation minimization in this work compares to Predictive Information?

---

### Official Review · Reviewer_RPqi · 2022-03-15

**Rating:** 2
**Confidence:** 1

**Review:**

This paper clearly contains enough material for a full conference paper, but the way it is compressed into 5 pages (and 13 appendix pages) makes it difficult to understand the key concepts / contributions. Many pages are spent on the background and the derivation of the loss function, while other essential parts of the method are described in one paragraph with a reference to an appendix. I believe a much better version of this paper could be made by highlighting the key concepts, succinctly describing the entire method, giving an intuitive explanation of the loss function, and delegating its derivation to the appendix. At the same time, I am sympathetic to the fact that the authors are dealing with a topic that is technically challenging.

The paper proposes a method for learning a model of a POMDP with the underlying state decomposed into factors. A key question the paper poses is what the minimum number of factors is to describe a particular environment.

The paper is definitely relevant to the workshop, but it is unclear to me if the main five pages can foster a good discussion. It definitely does not adhere to the following: "We ask authors to use the supplementary material only for minor details that do not fit in the main paper." I am uncertain about acceptance.

---

### Decision · Program_Chairs · 2022-03-24

Accept (Poster)